# Mutations in microRNA-128-2-3p identified with amplification-free hybridization assay

**Sofie Slott[1], Cecilie Schiøth Krüger-Jensen[1], Izabela Ferreira da Silva[2,3,4], Nadia Bom Pedersen[1], Kira Astakhova[ORCID][1] ***

**1** Department of Chemistry, Technical University of Denmark, Kgs Lyngby, Denmark, **2** Programa Interunidades de Pós-Graduacão em Bioinformática, Universidade Federal de Minas Gerais, Belo Horizonte-MG, Brazil, **3** Departamento de Física, Universidade Federal de Minas Gerais, Belo Horizonte-MG, Brazil, **4** Bioinformatics Core, Luxembourg Centre For Systems Biomedicine (LCSB), University of Luxembourg, Campus Belval, House of Biomedicine II, Belvaux, Luxembourg

* kirras@kemi.dtu.dk

**Data Availability Statement:** All relevant data are within the paper and its Supporting information files.

## Abstract

We describe a quantitative detection method for mutated microRNA in human plasma samples. Specific oligonucleotides designed from a Peyrard-Bishop model allowed accurate prediction of target:probe recognition affinity and specificity. Our amplification-free tandem bead-based hybridization assay had limit of detection of 2.2 pM. Thereby, the assay allowed identification of single-nucleotide polymorphism mismatch profiles in clinically relevant microRNA-128-2-3p, showing terminal mutations that correlate positively with inflammatory colitis and colorectal cancer.

## Introduction

Single-nucleotide polymorphisms (SNPs) are fundamental drivers of adaptation and disease origins in human cells. Therefore, SNPs can be used for patient stratification, facilitating personalized medicine approaches [1]. Detection of SNPs in human RNA is more challenging compared to DNA, as the RNA sequence is relative short, has a higher chemical and enzymatic instability, and the RNA active 3D folding decreases binding of the detection reagents [2].

In general, SNP detection reagents rely on matched probe:RNA oligonucleotide recognition, such as in sequencing and polymerase chain reaction (PCR). Single-cell RNA sequencing has recently been used to identify tumor-specific mutation pattern on expressed Acute Myeloid Leukemia cells [3]. However, sequencing is not suitable for large-scale screening, as next generation sequencing is limited by high cost and long assay time, while third generation sequencing does not meet specificity requirements [3,4]. PCR-restriction fragment length polymorphism (PCR-RFLP) has been used to identify SNPs in RNA. The method rely on PCR followed by enzymatic digestion to detect allelic variations by gel separation [5–7]. The major limitation of this method is the need to identify and apply only SNPs that overlap restriction enzyme recognition sites.

As alternative approach, mismatch hybridization strategies for SNP detection has been developed in previous studies for longer RNA targets [1,8]. *In situ* PCR in combination with FISH has also been developed to enable simultaneous detection of point mutations and copy

**Funding:** This study was financially supported by DTU Enable in the form of a grant (40996) awarded to KA. No additional external funding was received for this study. The funder had no role in study design, data collection and analysis, decision to publish, or preparation of the manuscript.

**Competing interests:** The authors have declared that no competing interests exist.

number variations at single-cell level in cancer tissue samples [9]. In this method, mutation and wild-type allele specific primers are applied for amplification, and then hybridization of fluorescently labeled probes specific for the wild-type and mutant PCR products are combined with other probes that enable copy number variation detection. MicroRNA, also abbreviated miRNA, are typically 20–22 nucleotide long nucleic acids that regulate expression of mRNA. MiRNAs have a confirmed role in multiple cancer types including breast, colon, prostate and lung cancers [10–14], through regulation of differentiation, apoptosis, migration and invasion [5,15–20]. Verified by next-generation sequencing, mutations in miRNA correlate with their biogenesis and are linked to inflammatory and cancerous diseases [15].

In light of the growing interest for SNP profiling in miRNAs, accurate and sensitive detection methods are on high demand [15]. In 2014, Deng et al. reported a miRNA detection assay called Toehold-Initiated Rolling Circle Amplification (TIRCA) to visualize miRNA *in situ* in single cells on single molecular level while being SNP sensitive [21].

Recently, Jia et al. applied a CRISPR/Cas12a RNA-directed nuclease system, in combination with hybridization chain circuit, for reverse transcription-free and amplification-free miRNA detection [22]. Although not sensitive to SNPs, the method could identify different miRNA in human cells in agreement with RT-qPCR, with a limit of detection down to 100 fM [22]. Moreover, as an effect of COVID-19 pandemic there has been an increased focus of rapid and efficient monitoring of mutations in viral RNA leading to improvements for nucleic acid detection. Zhang and Deng et al. recently describe a paper-based amplification-free assay (MARVE) for profiling Alpha, Beta and Gamma variants of SARS-Co2 in 100% agreement with RNAseq and RT-qPCR [23]. In previous work, we tested several miRNA in human biofluids (sera, plasma), and in rodent model of colitis-associated colon cancer, using a tandem hybridization bead-based assay [24,25]. We detected miRNA128-2-3p as potent biomarker for colon cancer progression. This miRNA was previously identified in several human tumors [26–28].

A potential pitfall and a scientific challenge of any miRNA detection method is the design of the specific detection probes. Recently, we applied a unique Peyrard-Bishop mesoscopic model to design oncogene-specific probes, which accounts for both hydrogen bonding and pi-stacking interactions between each individual nucleotide pair [29–31]. The model included locked nucleic acid (LNA) enriched probes that were designed specifically to target the desired DNA and RNA. LNA is a nucleic acid analogue which improves the probes' affinity and specificity especially for DNA:RNA recognition [24,29,32–34]. Using this model, we optimized oligonucleotide reagents so that the target SNP in coding RNA was discriminated with high specificity [31].

In this work, we hypothesized that an amplification-free assay with rationally designed LNA/DNA probes can be applied to rapid and accurate SNP profiling in human miRNA. We verified our hypothesis testing plasma sample using a bead-based fluorescence hybridization assay in a cohort of 24 colitis samples, 20 colorectal cancer (CRC) samples and 20 matched healthy controls.

## Materials and methods

### Mesoscopic model

The Peyrard-Bishop model is described in detail in SI. In brief, the model describes the DNA helix through a Hamiltonian, which contains a Morse potential describing the hydrogen bonds that connect each base-pair and a harmonic potential describing the stacking interaction of adjacent base-pairs [30,35–37]. Details on temperature prediction, parameter and DNA:RNA binding optimization are also provided in SI. The experimental dataset of melting

temperatures has 73 DNA:RNA hybrids sequences between 6-and 21-mer length at concentration of 100 mM [Na+] and 8 **μ**M of strand.

## Oligonucleotides

Biotinylated DNA and miR-128-2-3p were all purchased from IDT. The calf thymus DNA (CTD) binding linker was synthesized by solid-phase oligonucleotide synthesis. Details on linker synthesis and characterization is provided in Supporting Information.

## UV melting studies

Capture probes and model miRNA targets (see sequences in Table A in S3 File) were annealed by mixing equal molar amounts in 1x PBS, pH 7.4, at 2 μM final concentration, heating at 85˚C for 10 min, followed by cooling to room temperature over 1 h.

UV melting experiments were carried out on JASCO V-730 instrument equipped with PAC-743 Peltier 8-position cell changer. Each melting was done in duplicate; Tm was determined as second derivative of absorbance at 260 nm vs temperature curve.

## Fluorophore study in chip

For choosing the best dye for our miRNA detection assay, five different dyes were evaluated: EvaGreen (EG), AccuClear (AC), QuantiFluor (QF), Acridine Orange (AO) and Thiazole Orange (TO). Different dye concentrations were examined in order to investigate the intensity in presence of the CTD (booster). The samples were prepared by mixing 1xPBS with booster, and lastly the dye was added and mixed by vortex. The booster used was calf thymus DNA (CTD; Sigma reagent no. D4522) with stock concentration at 4 mg/ml. The samples were covered in aluminum foil to prevent exposure to UV-light. 45 μl of each sample was loaded in a one-chamber FLEET chip (design and production of the chips are described in SI) and analyzed by a Spectrofluorometer FS5 from Edinburgh Instruments with a SC-10 Front-Face Sample Holder module. Each test was run in duplicates. The CTD concentration in each sample was 2 μM (10 μL stock in 200 μL sample) and dye concentrations varied as followed: EG, AC, QF: 0.5%, 1%, 2.0%, 2.5%, 3.0%, 4.0% and AO, TO: 1.0%, 2.5%, 3.0%, 4.0%, 5.0%, 7.5%, 10.0%. Blanks had no CTD with 4% or 10% dye.

In order to test the photostability of the dyes, the samples were remeasured seven times. Each dye was tested in duplicates. After remeasurement, the photobleaching was calculated in percentage by the formula:

$$\% \, photobleaching = 100 - \left( \frac{\text{max last emission}}{\text{max first emission}} \cdot 100 \right)$$

To determine the limit of detection, we made a calibration curve for known amount of CTD with QuantiFluor fluorophore (CTD used in concentration range 0.1 pM-1 nM in presence of excess QuantiFluor fluorophore (100 nM), in $1 \times$ PBS, pH 7.2). The fluorescence read out was obtained using Roche Light Cycler 480 equipment, at excitation/emission wavelengths 480–510 nm/520-570 nm.

## Cohort information

De-identified human colitis plasma samples were obtained from Stanford University (USA) under Stanford Institutional Review Board (IRB protocol 28427). Human colon cancer (CRC) samples were provided by Danish Biobank Regionernes Bio- og GenomBank (RBGB) Herlev, under National Videnskabsetisk Komité (National Research Ethics Committee) (NVK)

number 21015520. All the samples were fully anonymized prior to analyses (S1 and S2 Tables). EDTA tubes were used for human blood collection. Plasma was obtained by 15 min centrifugation at 2,000 g in a bench top centrifuge (4˚C) of the fresh whole blood. The obtained plasma samples were stored in aliquots at -80˚C prior to usage. The participants were recruited for the study, their medical records and the samples were collected in the period 2005–2015. Analyses were carried out in the period 2018–2022. The authors had no access to any information that could identify the subjects neither during nor after the sampling completion.

### miRNA analysis

**Probe design.** The miRNA bead assay was conducted in duplicate following our published procedure [25]. The design principle of the probes has been described earlier [25]. miRNA target sequences were obtained from miR database (http://www.mirbase.org/). Complementary binary DNA probes were designed for each miRNA and enriched LNA to have high difference in melting temperature (Tm) for matched (mutated) vs mismatched (wild type) target:probe duplex. Resulting probe sequences are conserved for human and mice samples and are given as: miRNA128-2-3p target code: MIMAT0000424, wild type miRNA target: 5'-ucacagugaaccggucucuuu-3'.

The linker was synthesized by solid-phase oligonucleotide synthesis, purified, and characterized as described in SI. The CTD-binding part of the detection probes has been obtained from the National Center for Biotechnology Information (NCBI) Nucleotide database (sequence ID GJ060426.1).

**miRNA detection assay.** For miRNA detection, streptavidin magnetic beads (NEB S1420S; 10 µl, 4 mg/mL) were placed in an Eppendorf tube and the clear supernatant was discarded using a magnetic separation rack. The beads were then washed 3 times with 100 µl of 1 × PBS, pH 7.2. Hereafter, the magnetic beads were resuspended in 1× PBS (20 µl) and incubated with the miRNA-specific biotinylated capture probe (10 µl, 1 pmol) for 10 minutes on a shaker. When the incubation was finished the beads were washed with 1 × PBS (2 times by 100 µl) at RT and subsequently resuspended in 1 × PBS (20 µl).

Next, plasma sample (10 µl) or a known amount of synthetic miRNA target was added, and the resulting mixture was heated for 40 min at 65˚C with repetitive vortexing every 10 min, and subsequently cooled to 30˚C over 30 min followed by a wash with 1 × PBS (2 times by 100 µl) at 30˚C.

After cooling to RT, the CTD-signal boosting DNA (5 pmol; 2 µl) was added, followed by addition of the linker/detection probe (25 µl; 2 pmol), and 2 × PBS (25 µl) to the solution. The mixture was heated for 10 min at 85˚C and cooled to 16˚C over 20 min followed by a wash with 2 × PBS (3 times with 100 µl) at room temperature (23˚C). The mixture was resuspended with 2 × PBS (30 µl), heated 10 min at 92˚C and the supernatant was extracted. The supernatant was placed in a solution with QuantiFluor (3 µl, 20 × stock) and 2 × PBS (10 µl). The resulting mixture was vortexed for 7 min, followed by 5 min at 92˚C and finally cooled to room temperature (23˚C) over 60 min. Fluorescence read out was obtained using Roche Light Cycler 480 equipment, at excitation/emission wavelengths 480–510 nm/520-570 nm.

**Probe specificity.** Specificity of designed capture probes (C2-C13) were explored by comparing the read-outs from the assay using mixture of synthetic wild-type and mutant miRNA targets (0→100%). We performed the assay using mixtures of the synthetic wild type (M1) with synthetic mutant target (M2; M5; M8) of selected capture probes (C1, C2, C5, C8) (Fig A in S7 File). Total miRNA concentration was kept at 0.5 nM and the assay was performed as described above. Standard curve (S1 Fig) was applied to convert fluorescent signal to concentration plotted in Fig A in S7 File.

**Quantification and validation.** To quantify the amount of miRNA captured on the muta-tion-specific capture probe, we used a calibration curve (S1 Fig) from CTD in QF on the fluo-rescence readout obtained from the Roche Light Cycler 480. This was converted to the level of captured miRNA in each sample [nM] and are shown in Tables A-C in S5 File for CRC, colitis and healthy samples.

The results of the bead-based assay for wild type miRNA were verified by commercial Taq-Man RT-qPCR as described (ThermoFisher) [38]. DNase-treated RNA extracted from plasma samples (Qiagen RNeasy kit, as suggested by manufacturer), was retrotranscribed with the SuperScript™III First-Strand Synthesis System for RT-PCR (Invitrogen). RNA levels were ana-lyzed by real-time quantitative TaqMan RT-qPCR (ThermoFisher), following the manufactur-er's instructions. Reactions were performed with the LightCycler® 480 Instrument (Roche) in 384-multiwell plates. Primers for wild-type and mutated miR-128-2-3p (3' and 3'-1) were cus-tom designed and provided in a mastermix by ThermoFisher. Primers for second qPCR of selected mutant probes (3': U>G,C; 3'-1: U>G,C) were custom designed and provided in a mastermix by Genscript and run as described above. Thermal cycling was performed as fol-lows: one pre-incubation cycle at 95˚C for 10 min (ramp rate: 4.8˚C/s); 35 amplification cycles at 95˚C for 10 sec, 57˚C for 30 sec and 72˚C for 5 sec (ramp rates: 4.8, 2.2 and 4.8˚C/s). The obtained Ct values for detected miRNA (wt, 3' and 3'-1 variants) were converted to concentra-tions by use of the standard curve (Fig A in S5 File).

## Statistical analysis

Statistical data analysis was performed by applying descriptive statistics, Students t test, Welch test and regression analyses Pearson correlation, linear regression), in R. A P-value $\leq$ 0.05 was considered statistically significant.

## Results

### Fluorophore optimization

This study started by developing an ultra-sensitive amplification-free protocol that would allow detection of miRNA variants at low concentration. We choose the tandem hybridization bead-based method, used in previous work, and optimized the protocol to get lower limit of detection (LOD) [24,25]. Calf thymus DNA (CTD) was used as a scaffold for multiplying the number of fluorophores per target RNA:probe binding event [25].

We selected a chip format for rapid screening of different dyes compatible with hybridiza-tion-based nucleic acid detection. The chip was prepared as a one-chamber device made of PMMA (described in Supporting Information). Using this convenient device, we rapidly investigated five fluorescent dyes that recognize DNA; EvaGreen, AccuClear, QuantiFluor, Acridine Orange and Thiazole Orange for photostability and emission intensity in the pres-ence of CTD used as signal booster. Fig 1A shows representative emission spectrum of Quanti-Flour. As shown in Fig 1B, QuantiFluor and Thiazole Orange emit the most florescence relative to their concentration.

Photobleaching happens over multiple absorption- and emission-circles, where a fluoro-phore loses the ability to emit light [39]. Photobleaching was examined by remeasuring the same samples seven times in duplicates. Fig 1C shows the percentage photobleaching observed a different dye concentration. AccuClear and QuantiFluor were found to be most photostable.

Based on the data from the emission and photobleaching studies, we selected QuantiFluor as the "booster" dye for our miRNA detection assay, giving a limit of detected (LOD) of 2.2 pM and bleaching 2.5% on average over seven exposures.

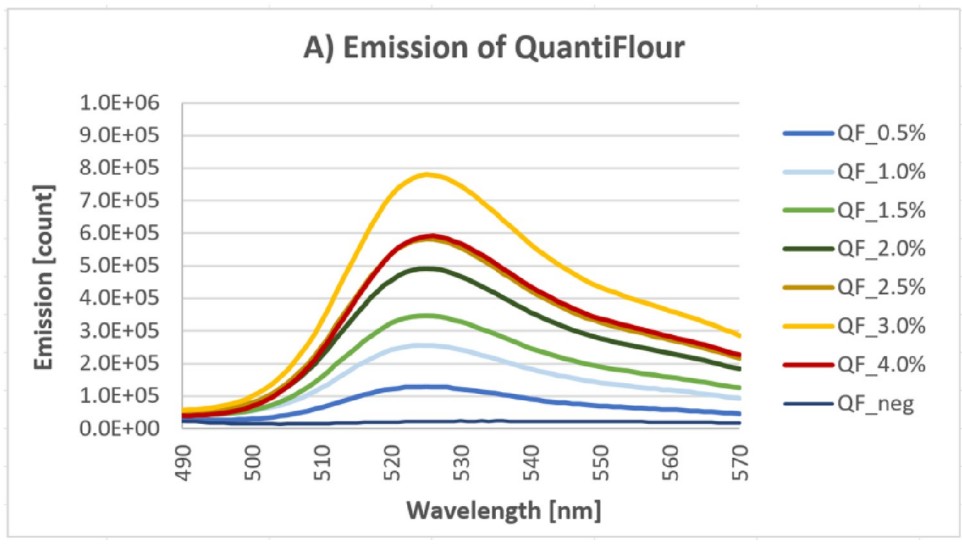

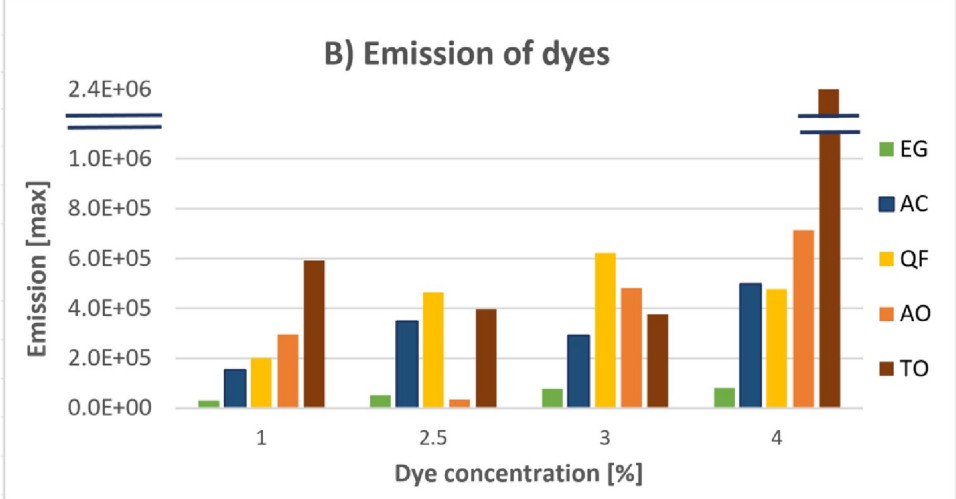

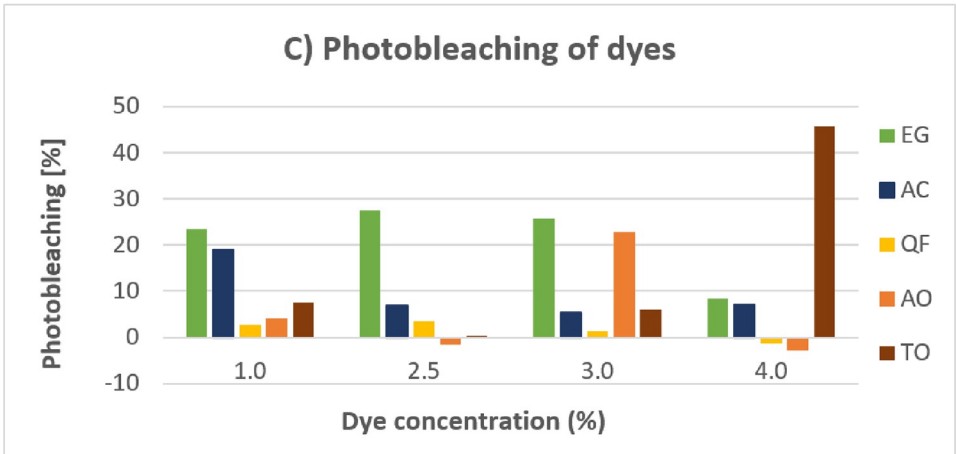

**Fig 1. Fluorophore study in chip.** Five dyes were evaluated, namely EvaGreen (EG), AccuClear (AC), QuantiFluor (QF), Acridine Orange (AO) and Thiazole Orange (TO). A) Representation emission spectrum of QF. B) Grouped plot of the maximum emission values for each dye in comparable concentrations. C) Photobleaching of dyes over 7 measurements (% from first to last measurement).

## SNP probe design

After determining the optimal booster dye, we proceeded with analysis of miRNA in human plasma samples. For this, we selected miRNA128-2-3p as a potent biomarker with upregulation in colitis and colorectal cancer [24].

The probes for the assay were designed as a tandem hybridization system, where biotinylated capture probe attached to the magnetic bead surface was a mutation-specific LNA-enriched oligonucleotide (Fig 2). Secondary probe, called linker, binds to the 5' end of the miRNA and contains sequence that specifically interacts with the signal booster. It has sequence: 5'-T+C+A+CT+GT+GATTTTGAT+GGG+AATAC+CAGACC-3'. The first part (dashed) adjusts to 5'-end of the target miRNA with a predicted Tm of 45 C°. The underlined 3'–end binds to CTD. LNA nucleotides are indicated with a plus (+) in front of the corresponding nucleotide letter. As a proof of principle, we designed 12 mutation specific capture probes of 12 nucleotides in length, recognizing mutations at 3' terminus and 3 adjacent positions (3', 3'-1, 3'-2, 3'-3) of the target miRNA.

The probe sequences were designed using a parameterization of the Peyrard-Bishop model for LNA-modified probes. The model describes the sequences through the hydrogen-bonds between each base-pair and their stacking interaction [30,31]. The design allowed up to four LNA additions per sequence to reach improved binding affinity to target miRNA. We aimed at a drop in the melting temperature in the presence of mismatch in the target. Final probes and their predicted discrimination of mismatch ($\Delta T_m$) are shown in Table 1. The design was successful in terms of discriminating SNP; with melting temperature difference ($\Delta T_m$) for match vs. mismatch in range 8.8°C– 17.2°C (Table 1).

**Tandem hybridization bead-based assay—Detecting SNPs in miR-128-2-3p.** Prior to the assay, melting temperatures of capture probes were evaluated by a thermal denaturation study. We analyzed C1-C7 with complement model miRNA targets (M1-M7, Table A in S3 File) and results are seen in Table B in S3 File. Difference in Tm between two independent measurements did not exceed 1.5°C. The experimental $\Delta Tm$ of matched vs. mutant was found to range between 6.6°C– 25.5°C (Table 1, $\Delta Tm$ exp.), which is a slightly broader range than predicted. However, these UV melting temperature studies confirmed that the designed capture probes could effectively discriminate between the wild-type miRNA128-2-3p and its mutated analogues.

For the amplification-free assay, biotinylated capture probes C1-C14 were attached to streptavidin-coated magnetic beads in individual reaction tubes (Fig 3). We used elevated temperature (30°C) in the following washing step to ensure that unspecific capture probe:target binding did not take place. After sequential incubation with patient plasma sample, linker probe and booster were added. Followed by several washes, the complex was denatured, and

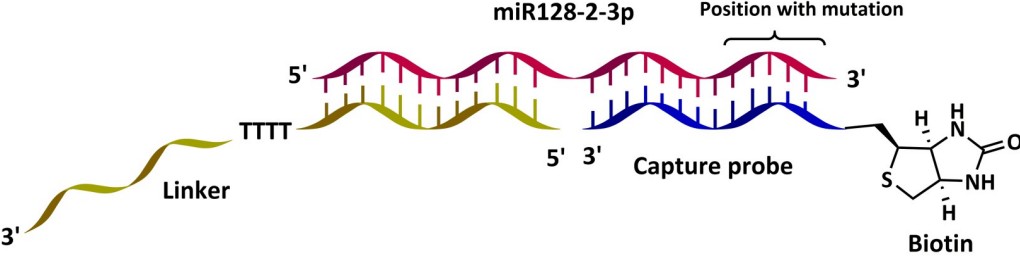

**Fig 2. Design of tandem hybridization probes for this study.** Biotinylated mutation specific capture probe (blue), miRNA128-2-3p or variant (pink), linker (yellow) with booster binding part. CTD = calf thymus DNA (booster).

**Table 1. Mutation specific capture DNA/LNA probes targeting miRNA128-2-3p.**

| # | Sequence, 5′->3′ | Position of Mutation | $\Delta T_m$ pred, ˚C | $\Delta T_m$ exp, ˚C |
|---|---|---|---|---|
| C1 | Bt-Sp9-d(AAAGAG+AC+CG+GT) | None, wt | 13.3 | 17.3 |
| C2 | Bt-Sp9-d(TAAGAG+AC+CG+GT) | 3', U>A | 11.3 | 16.4 |
| C3 | Bt-Sp9-d(CAAGAG+AC+CG+GT) | 3', U>G | 11.3 | 12.8 |
| C4 | Bt-Sp9-d(GAAGAG+AC+CG+GT) | 3', U>C | 11.3 | 6.6 |
| C5 | Bt-Sp9-d(ATAGA+GACC+G+GT) | n-1, U>A | 10.0 | 18.4 |
| C6 | Bt-Sp9-d(ACAGA+GACC+G+GT) | n-1, U>G | 10.0 | 10.2 |
| C7 | Bt-Sp9-d(+AGA+GAGA+CCGGT) | n-1, U>C | 10.3 | 25.5 |
| C8 | Bt-Sp9-d(AAT+GAG+A+CCGGT) | n-2, U>A | 10.8 | NM |
| C9 | Bt-Sp9-d(AAC+GAGA+CCGGT) | n-2, U>G | 10.8 | NM |
| C10 | Bt-Sp9-d(AAG+GAGA+CCGGT) | n-2, U>C | 10.8 | NM |
| C11 | Bt-Sp9-d(AAA+T+AGACC+GGT) | n-3, C>A | 8.8 | NM |
| C12 | Bt-Sp9-d(AAACAGA+CCG+GT) | n-3, C>G | 17.2 | NM |
| C13 | Bt-Sp9-d(AAAAAG+A+CCG+GT) | n-3, C>U | 14.6 | NM |
| C14 | Bt-Sp9-d(AGCGCGGATAAA) | Scramble | 14.6 | NM |

Probes applied in this study; predicted and experimental difference in melting temperatures ($\Delta T_m$) between fully matched (mutant) and mismatched (wild type) RNA: capture probe complexes; Bt = Biotin; Sp9 = spacer9; underlined = mutation; '+' = LNA modification.

the supernatant transferred to a QuantiFluor solution after which the fluorescence signal was detected (Fig 3).

A fluorescence calibration curve was established to convert optical read out of the amplification-free assay into absolute amounts of either wild-type or mutated miRNA. According to our design, there is a large difference in affinity for wild type and mismatch probe (Table 1). However, there is a possibility of unspecific binding of wild-type (miRNA128-2-3p) on mutant capture probes (C2-C13) leading to false positive readouts. To explore specificity of designed probes to mutant miRNA, we performed analyses for mixed wild-type and mutated miRNA as synthetic targets in the bead assay (Fig A in S7 File). The result confirms high specificity of the probes, with a consistent drop of the signal at increased presence of the mismatched (wild type) miRNA. The absolute amounts of wild-type and mutated miRNA obtained for human plasma samples are shown in Fig 4. For the wild-type target, the assay was validated using RT-qPCR as we reported before [25]. Regression analysis gave the following model of the concentration of miRNA128-2-3p in nM for all tested samples: c(bead) = 0.49*c(qPCR), p-value = $3.2*10^{-9}$, $r^2 = 0$, where bead refers to our bead-based hybridization assay. The model reveals a higher (p-value = $7.7*10^{-4}$) wild-type miRNA titer determined by RT-qPCR.

According to one-way ANOVA, there is a statistically significant difference among all groups in diseased samples (Fig 4, green and orange), and not in healthy controls (Fig 4, grey). Notably, levels of mutated miRNA are higher in the diseased samples than healthy controls. It indicates an association between SNP mutations in miR-128-2-3p and diseases colitis and colorectal cancer. Next, there is a high similarity in miRNA mutation pattern between colitis and colorectal cancer, with slightly higher levels of mutated miRNA in the former. In both diseased groups, levels of terminal mutated miR-128-2-3p in the first three positions (C2-C10) are higher than for healthy controls. Mutations 3'-2 U>G and 3'-2 U>C give the highest levels. Opposite, for 3'-3 position the levels of mutated miRNA128-2-3p are lowest. Using a QuantiFluor calibration curve, LOD of the assay is determined to be 2.2 pM RNA target which is 5 times lower compared to previously used EvaGreen [24,25]. High specificity of the assay is

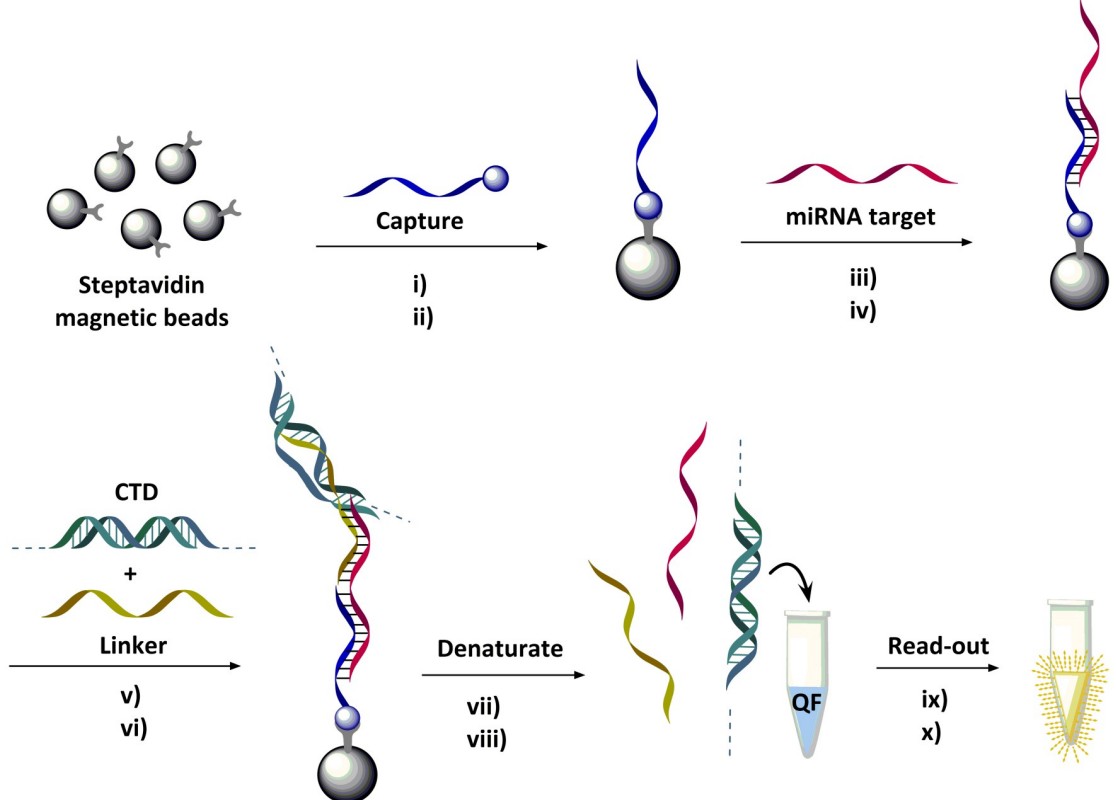

**Fig 3.** Procedure of miRNA detection bead-based assay: i) Streptavidin magnetic beads were incubated with biotinylated mutation specific capture probes (C1-C14) followed by ii) washing, iii) incubation with samples containing miRNA128-2-3p and variants then iv) washing. v) Beads were then incubated with CTD, linker and buffer followed by vi) washing and vii) denaturation by heating to 92°C in 10 min, and viii) the supernatant transferred to QuantiFluor (QF) solution, ix) that was incubated at 92°C in 5 min and cooled to 23°C over 60 min. Finally, x) the concentration of miRNA128-2-3p and variants were quantified by fluorescence detection.

confirmed with no binding for scrambled control probe C14 and verification of selective mutation binding for capture probes (Fig A and Table A in S7 File).

## Correlation with other clinical parameters

To further explore the biological role of the SNP mutations in miRNA128-2-3p, we performed Pearson correlations of clinical features for subjects with obtained levels of miRNA (Table B in S6 File). Relatively weak correlations were observed, with Pearson coefficients in the range -0.34 to +0.38. Interestingly, multiple correlations were observed for same few miRNA variants, namely C4, C6, C7 and C10. Age correlated negatively with C4 and C12 levels (Pearson coefficient -0,21 and -0.3, respectively); as to gender, there were no correlations. Disease type and stage correlated negatively with levels of several mutated miRNAs, also with wild type (C1, C4 and C10). A positive correlation was seen for history of ulcerative colitis with C8 (Pearson coefficient 0.38); and for hemoglobin levels correlating positively with C4 and C10 (Pearson coefficient 0.29 and 0.27, respectively), and erythrocyte sedimentation rate (ESR), correlating positively with C5 (Pearson coefficient 0.21). Last, we observed positive correlation of mutated miRNA128-2-3p (C6, C7 and C13), with race of subjects, no-White group having higher correlation with the mutated miRNA.

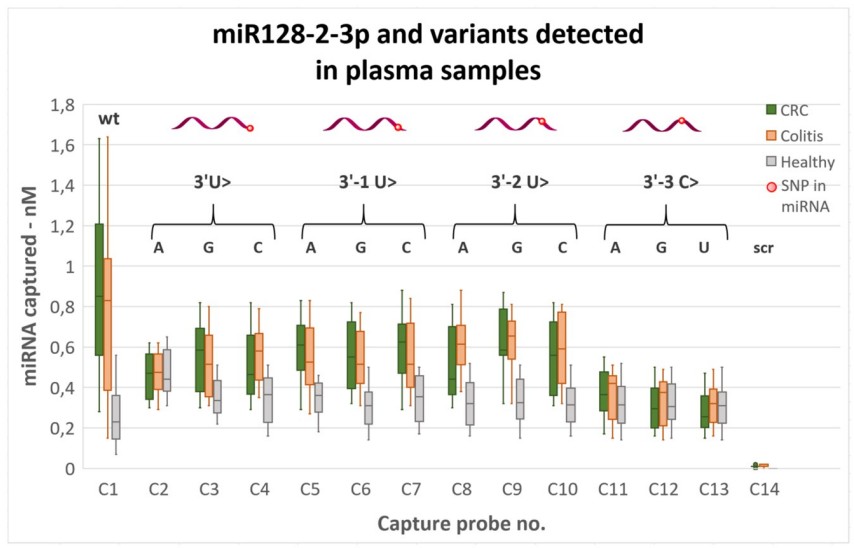

**Fig 4. Results of amplification-free hybridization assay in human colorectal cancer, colitis and healthy plasma samples.** miRNA readout in colorectal cancer (green), colitis (orange) and healthy (grey, ns all groups) samples using amplification-free assay in box-whisker plot with outliers. miRNA targets are illustrated with position of mutation (red dot). Letters show the individual modifications, correlating with the capture probe numbers (C2-C13); wt = wild-type (C1); scr = scrambled capture probe (C14). ns = not significant (p > 0.05).

## Discussion

In this study, we optimize a bead-based hybridization assay to accurately detect SNPs in human miRNA128-2-3p. From examinations of five different dyes, QuantiFluor was chosen for the miRNA detection bead-based assay as it had the best emission qualities and photo-stability. By this, we achieved an optimized LOD from 10pM to 2.2pM [25].

In the literature, the majority of existing miRNA detection methods require enzymes for signal amplification such as RT-PCR, rolling circle amplification, strand displacement amplification and exonuclease/nicking endonuclease-assistant amplification [40]. Only few amplification-free methods sensitive to miRNA SNPs have been developed, and even less has been applied to detect SNPs in miRNA [40–42]. In 2014, Qiu et al. described the use of a hybridization chain reaction to fluorescently identify let-7a on SNP level using two different silver nanocluster hairpin oligomers [40]. Caputo et al. (2019) designed a miRNA detection platform using microgels and molecular beacon stem loops, that when conjugated together, could catch miRNA in samples [41]. The biosensor showed SNP selectivity and low sensitivity (nanomolar to picomolar order, LOD 10 fM, detection time 1h). The miRNA detection methods described in Caputo et al. and Qiu et al. both show SNP selectivity. However, the methods is not applied for detection of specific miRNAs SNPs in the biological samples and are only analyzing the wild-type miRNA [40,41]. A recent study by Xia et al. [43] applied the SYBR green fluorophore for SNP detection. In the assay, graphene oxide was added to prevent SYBR green dsDNA insertion, quenching the fluorescent signal and thus allowing SNP detection. The method has LOD of 1nM, but was used for DNA rather than (mi)RNA targets, which were synthetic rather than biological [43].

It is well known that mutations appear at an extremely low frequency in genome and transcriptome [44,45]. Although amplification-free approaches can efficiently reduce the complexity of an experimental system, the detection sensitivity of these assays can in general not meet

the requirements of biological analysis and clinical applications. In this assay, the detection limit is determined to be 2.2 pM, which is poorer than isothermal exponential amplification reaction (EXPAR) with chimeric DNA probe-aided precise RNA disconnection (1fM) [46] and tandem gene amplification (8.3 fg) [47]. Nevertheless, given relatively high miRNA concentration in the samples we tested (0.07–1.64 nM), applying our method is feasible (Supplementary Tables A-C in S5 File).

Using our probes in an amplification-free assay, we show that wild-type miRNA128-2-3p is present in a significantly higher amount in colitis and CRC patients compared to healthy controls and therefore, the miRNA can be used as a biomarker for colon cancer progression. Furthermore, we were able to quantify the amount of SNP variants in several positions of miRNA128-2-3p obtained from the samples from colitis and CRC patients. To the best of our knowledge, SNP in miRNA128-2-3p associated with colitis or CRC have not yet been reported.

A risk of false positive signal is considerable when detecting miRNA SNP such as with miR-141 and miR-200a reported by Metcalf et al. [42]. However, our data can be fitted with TaqMan-qPCR data, confirming accuracy of the developed assay. For the RT-qPCR, a higher amount of wild-type miRNA128-2-3p was detected compared to our bead-based assay, which lowers the risk of false positive samples. However, it also indicates that the calibration and accuracy of the assay can be optimized further.

Applying our bead-based assay in human plasma samples, we detect a significant difference in mutated miRNA levels of diseased samples, but not in healthy controls. It indicates that healthy individuals have baseline levels of SNP miRNA128-2-3p, some of which gets elevated upon disease state. Therefore, the miRNA128-2-3p SNPs giving the best discrimination of healthy and disease individuals can be selected as biomarkers of CRC and colitis. Our study indicates that SNP mutations 3'-2U>G and 3'-2U>C (capture probes C9 and C10) are promising biomarkers of CRC (median difference is 44% for both, P-values $3.1*10^{-8}$ and $1.2*10^{-5}$, respectively). Likewise, SNP mutations 3'-2U>A and 3'-2U>G (capture probes C8 and C9) discriminates colitis well which median differences of 48% (p-value: $1.6*10^{-9}$) and 50% (p-value: $1.6*10^{-8}$), respectively. In addition, Person correlation was positive between 3'-2U>A (C8) and history of ulcerative colitis (Pearson coefficient: 0.38), while 3'-2U>C (C10) positively correlated with hemoglobin (Pearson coefficient: 0.27), while negatively with disease type and stage (Pearson coefficient: -0.23). Importantly, the 3'-2 mutations were not correlated with either age or gender. Taken together, 3'-2 SNP mutations of miRNA128-2-3p are promising as biomarkers of CRC and colitis. However, the finding should be verified in a larger cohort.

This work demonstrates the utility of a tandem hybridization bead-based assay to detect SNP mismatch profiles specifically in miRNA128-2-3p. However, we predict a potential to expand the technology to detection of various clinical and biological relevant RNA biomarkers. The developed Peyrard-Bishop model offers a tool for design of high-sensitivity probes, which can adapt the assay to screening of any miRNA SNP, potentially allowing early disease diagnosis.

## Conclusion

In summary, we designed specific hybridization probes to detect SNPs in miRNA128-2-3p. We applied the Peyrard-Bishop mesoscopic model to design LNA enriched capture probes with high specificity for miRNA128-2-3p and its mutants. We showed high discrimination between match and mismatch probes with a $\Delta T_m$ range between 6.6–25.5°C. The mutation-specific capture probes were incorporated into bead-based amplification-free detection

platform. Using the assay, we successfully quantified levels of miRNA128-2-3p and its mutated variants in plasma samples from colitis and colorectal cancer patients. We improved the LOD from 10 pM to 2.2 pM using the fluorescent dye QuantiFlour. Using the bead-assay we demonstrated a significantly higher level of the wild-type miRNA128-2-3p in colitis and colorectal cancer than in healthy. Moreover, we observed a generally higher amount of mutated miRNA128-2-3p in colitis and CRC than in healthy controls, except for mutations in the 3'-3 position. Our approach can be used to detect various biologically and clinically relevant RNA biomarkers, which might be useful in fast and efficient early stage diagnosing or screening.

## Supporting information

**S1 Fig. Calibration curve for CTD using QuantiFluor dye.** The LOD with signal to noise ratio > 3, was determined to be 2.2 pM for QuantiFluor, based on the calibration curves. (TIF)

**S1 Table. Clinical and demographical information of CRC patients.** De-identified human colon cancer samples, along with 10 healthy controls (gender: 5 male and 5 female; Median age at sample (range) 44 (30–53)) were provided by Danish Biobank Regionernes Bio- og GenomBank (RBGB) Herlev, under National Videnskabsetisk Komité (National Research Ethics Committee) (NVK) number 21015520. (DOCX)

**S2 Table. Clinical and demographical information of inflammatory bowel disease patients.** De-identified human colitis plasma samples, along with 10 healthy controls (gender: 5 male and 5 female; Median age at sample 44), were obtained from Stanford University (USA) under Stanford Institutional Review Board (IRB protocol 28427). (DOCX)

**S1 File. Mesoscopic modelling and parameter optimization of DNA:RNA hybrids.** Mesoscopic model, temperature predictions, parameter optimizations, DNA:RNA optimization and experimental data. (DOCX)

**S2 File. Oligonucleotides.** Sequences of purchased oligonucleotides, synthesized oligonucleotide, characterization and purification. (DOCX)

**S3 File. UV melting studies of designed DNA:RNA hybrids.** Sequences of model miRNA targets used in UV thermal denaturation study; Tm results for duplexes of biotinylated capture probes C1-C7 with complementary and mismatched model miRNA targets; Representative UV melting curves for capture probe. (DOCX)

**S4 File. Dye study in chip to optimize amplification-free assay.** Chip design and production; fluorophore and photobleaching study in chip. (DOCX)

**S5 File. miRNA analysis using amplification-free bead assay and RT-qPCR.** Calibration curve for RT-qPCR for wild type analyses (miRNA128-2-3p); Concentrations of miRNA128-2-3p and its variants detected in plasma of patients with CRC, colitis and healthy controls using bead-assay and qPCR; Mean CV% for commercial and in house RT-qPCR of miRNA128-2-3p and its mutated variants in position 3' and 3'-1. (DOCX)

**S6 File. Statistical analyses.** Linear regression study for amplification-free bead assay vs. qPCR; Pearson coefficients for clinical correlation of amplification-free assay.
(DOCX)

**S7 File. Validation of probe specificity.** Probe specificity for mixed synthetic target miRNA; Concentrations of miRNA128-variants (M3, M4, M5, M6) detected in plasma of patients with CRC using qPCR and bead assay; Linear regression for amplification-free bead assay vs. qPCR.
(DOCX)

## Acknowledgments

We thank Stanford University, School of Medicine, CA, USA, and Danish Biobank Regionernes Bio-og GenomBank (RBGB) Herlev, for providing us with colitis, colon cancer plasma samples and healthy controls.

## Author Contributions

**Conceptualization:** Sofie Slott, Cecilie Schiøth Krüger-Jensen, Izabela Ferreira da Silva, Nadia Bom Pedersen, Kira Astakhova.

**Data curation:** Sofie Slott, Cecilie Schiøth Krüger-Jensen, Izabela Ferreira da Silva, Nadia Bom Pedersen, Kira Astakhova.

**Formal analysis:** Sofie Slott, Cecilie Schiøth Krüger-Jensen, Izabela Ferreira da Silva, Nadia Bom Pedersen, Kira Astakhova.

**Investigation:** Sofie Slott, Cecilie Schiøth Krüger-Jensen, Kira Astakhova.

**Methodology:** Sofie Slott, Cecilie Schiøth Krüger-Jensen, Izabela Ferreira da Silva, Kira Astakhova.

**Project administration:** Kira Astakhova.

**Resources:** Kira Astakhova.

**Software:** Izabela Ferreira da Silva.

**Supervision:** Kira Astakhova.

**Validation:** Izabela Ferreira da Silva, Nadia Bom Pedersen, Kira Astakhova.

**Visualization:** Sofie Slott, Cecilie Schiøth Krüger-Jensen, Izabela Ferreira da Silva, Nadia Bom Pedersen, Kira Astakhova.

**Writing – original draft:** Sofie Slott, Cecilie Schiøth Krüger-Jensen, Izabela Ferreira da Silva, Nadia Bom Pedersen, Kira Astakhova.

**Writing – review & editing:** Sofie Slott, Cecilie Schiøth Krüger-Jensen, Izabela Ferreira da Silva, Nadia Bom Pedersen, Kira Astakhova.

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
