## [Decision Letter · Decision Letter 0]

23 May 2023

PONE-D-23-09799Mutations in microRNA-128-2-3p identified with amplification-free hybridization assayPLOS ONE

Dear Dr. Astakhova,

Thank you for submitting your manuscript to PLOS ONE. After careful consideration, we feel that it has merit but does not fully meet PLOS ONE’s publication criteria as it currently stands. Therefore, we invite you to submit a revised version of the manuscript that addresses the points raised during the review process.

We look forward to receiving your revised manuscript.

Kind regards,

Ruslan Kalendar

Academic Editor

PLOS ONE

Reviewers' comments:

Reviewer's Responses to Questions

**Comments to the Author**

1. Is the manuscript technically sound, and do the data support the conclusions?

Reviewer #1: Yes

Reviewer #2: Partly

2. Has the statistical analysis been performed appropriately and rigorously? 

Reviewer #1: Yes

Reviewer #2: No

3. Have the authors made all data underlying the findings in their manuscript fully available?

Reviewer #1: Yes

Reviewer #2: Yes

4. Is the manuscript presented in an intelligible fashion and written in standard English?

Reviewer #1: Yes

Reviewer #2: No

5. Review Comments to the Author

Reviewer #1: 

The manuscript proposed a quantitative detection method for mutated microRNA in human plasma samples. The authors designed an amplification-free assay with rationally designed LNA/DNA probes that can be applied to profile SNP. This manuscript is reasonably organized and the assay has potentials to be used for diagnosing cancer cells. Some issues should be addressed before published:

1. The introduction is a bit lengthy, and it is suggested that can be simplified. For example, the authors can emphasize the advantages of this proposed method compared with conventional assays.

2. The proposed model included locked nucleic acid (LNA) enriched probes that were designed specifically to target RNA. The author indicated that LNA is a nucleic acid analogue which improves the probes’ affinity and specificity. Some experimental data need to be presented to prove it.

3. The manuscript indicated that amorphous Mn-containing catalysts are featured by mixed Mn oxidation states (III/IV) and disordered Mn geometry. What are the catalytic effects of different valence states of Mn ions?

4. The resolution of figures should be improved to show clearly, and the axes inside each figure can be enlarged to show clearly.

5. Some assays for SNPs analysis should be introduced, such as Nature Biomedical Engineering, 2022, 6, 957-967; Nature Communications, 2019, 10, 3660; Nature Genetics, 2015, 47, 1212-1219; Angewandte Chemie International Edition, 2014, 53, 2389-2393.

Reviewer #2: 

In the manuscript entitled “Mutations in mircroRNA-128-2-3p identified with amplification-free hybridization assay”, the authors used a quantitative method to detect mutated miR-128-2-3p SNP in inflammatory colitis and colorectal cancer samples. The study emphasizes the significance of SNP detection in mutant miRNA within patient samples. However, the manuscript exhibits readability issues, including grammar and spelling errors, and lacks appropriate citations. To enhance the manuscript further, the following points should be addressed:

Major Points:

1. The designed C2-C13 DNA/LNA probe may still base pair with any variant of miR-128-2-3p due to the 11-nt base pairing that remains after binding with LNA at 30˚C, leading to a high incidence of false positive reads. To address this issue, the authors can perform in vitro or cell-free assays to validate and quantify the false positive reading bias in their methodology.

2. To establish greater confidence in the application of this method, the authors can employ small RNA sequencing (small RNA-seq) to verify the endogenous expression levels of miRNA-128-2-3p mutants in samples from patients with inflammatory colitis and colorectal cancer.

Minor Points:

1. In line 26, "accroaches" should be corrected to "approaches."

2. In lines 134-135, "Then the beads were washed 3 times using each time 100µl 1X PBS, pH 7.2" can be revised to "The beads were then washed 3 times with 100µl of 1X PBS, pH 7.2."

3. In line 307, "However, is" should be changed to "However, it."

4. In line 281, the first author of the citation "Qui et al" should be corrected to "Qiu."

5. To enhance readability, the authors can provide high-resolution figures.

6. PLOS authors have the option to publish the peer review history of their article (what does this mean?). If published, this will include your full peer review and any attached files.

Reviewer #1: **Yes: **Ruijie Deng

Reviewer #2: No

---

## [Author Response · Author response to Decision Letter 0]

6 Jul 2023

Dear Referees and Editor

Thank you very much for your comments which we found useful for improving our manuscript.

Implemented changes are listed in the uploaded response letter and are highlighted in the revised version of the paper and Supporting information.

---

## [Decision Letter · Decision Letter 1]

21 Jul 2023

Mutations in microRNA-128-2-3p identified with amplification-free hybridization assay

PONE-D-23-09799R1

Dear Dr. Astakhova,

We’re pleased to inform you that your manuscript has been judged scientifically suitable for publication and will be formally accepted for publication once it meets all outstanding technical requirements.

Kind regards,

Ruslan Kalendar

Academic Editor

PLOS ONE

<quillbot-extension-portal></quillbot-extension-portal>

---

## [Editor Report · Acceptance letter]

10 Aug 2023

PONE-D-23-09799R1 

Mutations in microRNA-128-2-3p identified with amplification-free hybridization assay 

Dear Dr. Astakhova:

I'm pleased to inform you that your manuscript has been deemed suitable for publication in PLOS ONE. Congratulations! Your manuscript is now with our production department. 

Kind regards, 

on behalf of

Professor Ruslan Kalendar 

Academic Editor

PLOS ONE